# From Recognition to Remedy: The Significance of Biomarkers in Neurodegenerative Disease Pathology

**DOI:** 10.3390/ijms242216119

**Published:** 2023-11-09

**Authors:** Corneliu Toader, Nicolaie Dobrin, Felix-Mircea Brehar, Constantin Popa, Razvan-Adrian Covache-Busuioc, Luca Andrei Glavan, Horia Petre Costin, Bogdan-Gabriel Bratu, Antonio Daniel Corlatescu, Andrei Adrian Popa, Alexandru Vlad Ciurea

**Affiliations:** 1Department of Neurosurgery, “Carol Davila” University of Medicine and Pharmacy, 020021 Bucharest, Romania; corneliu.toader@umfcd.ro (C.T.); luca-andrei.glavan0720@stud.umfcd.ro (L.A.G.); horia-petre.costin0720@stud.umfcd.ro (H.P.C.); bogdan.bratu@stud.umfcd.ro (B.-G.B.); antonio.corlatescu0920@stud.umfcd.ro (A.D.C.); prof.avciurea@gmail.com (A.V.C.); 2Department of Vascular Neurosurgery, National Institute of Neurology and Neurovascular Diseases, 077160 Bucharest, Romania; 3Department of Neurosurgery, Clinical Emergency Hospital “Prof. Dr. Nicolae Oblu”, 700309 Iasi, Romania; 4Department of Neurosurgery, Clinical Emergency Hospital “Bagdasar-Arseni”, 041915 Bucharest, Romania; 5Department of Neurology, “Carol Davila” University of Medicine and Pharmacy, 020021 Bucharest, Romania; 6Department of Neurology, National Institute of Neurology and Neurovascular Diseases, 077160 Bucharest, Romania; 7Medical Science Section, Romanian Academy, 060021 Bucharest, Romania; 8Neurosurgery Department, Sanador Clinical Hospital, 010991 Bucharest, Romania

**Keywords:** neurodegenerative diseases, aging population, socio-economic implications, Alzheimer’s disease, Parkinson’s disease, amyotrophic lateral sclerosis, biomarker identification, wet and dry markers, early diagnosis, disease monitoring, therapeutic efficacy, susceptibility markers, trait markers, personalized medicine, diagnostic accuracy, disease progression, innovative techniques, -omics technologies, histologic assessments, imaging technology

## Abstract

With the inexorable aging of the global populace, neurodegenerative diseases (NDs) like Alzheimer’s disease (AD), Parkinson’s disease (PD), and amyotrophic lateral sclerosis (ALS) pose escalating challenges, which are underscored by their socioeconomic repercussions. A pivotal aspect in addressing these challenges lies in the elucidation and application of biomarkers for timely diagnosis, vigilant monitoring, and effective treatment modalities. This review delineates the quintessence of biomarkers in the realm of NDs, elucidating various classifications and their indispensable roles. Particularly, the quest for novel biomarkers in AD, transcending traditional markers in PD, and the frontier of biomarker research in ALS are scrutinized. Emergent susceptibility and trait markers herald a new era of personalized medicine, promising enhanced treatment initiation especially in cases of SOD1-ALS. The discourse extends to diagnostic and state markers, revolutionizing early detection and monitoring, alongside progression markers that unveil the trajectory of NDs, propelling forward the potential for tailored interventions. The synergy between burgeoning technologies and innovative techniques like -omics, histologic assessments, and imaging is spotlighted, underscoring their pivotal roles in biomarker discovery. Reflecting on the progress hitherto, the review underscores the exigent need for multidisciplinary collaborations to surmount the challenges ahead, accelerate biomarker discovery, and herald a new epoch of understanding and managing NDs. Through a panoramic lens, this article endeavors to provide a comprehensive insight into the burgeoning field of biomarkers in NDs, spotlighting the promise they hold in transforming the diagnostic landscape, enhancing disease management, and illuminating the pathway toward efficacious therapeutic interventions.

## 1. Introduction

### 1.1. The Growing Challenge of Neurodegenerative Diseases Due to the Aging Population

Given the recent shifts in global demography, there has been a significant rise in the older adult cohort, specifically those above 65 years. Contemporary demographic studies indicate that this age group is expanding more rapidly than its counterparts. Data from 2019 show that 703 million individuals globally were aged 65 or older, which is a figure that is projected to surge to 1.5 billion by 2050 [1]. In a 2018 study by the American Academy of Neurology, it was inferred that between 15 and 20% of those aged 65 and above exhibited symptoms of mild cognitive impairment (MCI), which is a syndrome characterized by subtle cognitive decline with negligible impact on daily functional activities [2]. The escalating aging trend worldwide is concomitant with the rise of age-related health challenges. Notably, recent analyses have emphasized that neurological disorders stand as the primary contributors to DALYs (disability-adjusted life-years), accounting for 276 million cases, and are the second predominant cause of mortality, with 90 million cases [3]. Corroborating this, there is compelling epidemiological data suggesting a potential link between physical frailty and cognitive setbacks in older age, including the onset of Alzheimer’s disease (AD), MCI, vascular dementia (VaD), non-AD dementias, and the presence of AD pathology even in older individuals not diagnosed with dementia [4]. However, the current epidemiological landscape presents a scarcity of comprehensive data on cognitive frailty, particularly its prevalence and implications [5]. The trajectory from cognitive frailty to full-blown dementia remains ambiguous at present.

### 1.2. Understanding the Socioeconomic Implications of Increasing Prevalence of Conditions like AD, PD, and ALS

Addressing the economic repercussions due to the upsurge in Alzheimer’s disease is paramount. A proactive approach involving early detection and intervention is vital not only to mitigate the prevalence of AD but also to elevate the life quality of both the affected individuals and their caregivers. The institution of robust social support mechanisms is integral to this strategy. Non-pharmacological measures emerge as the most preferred modalities in both the prevention and management of AD [6]. There is a prevailing hypothesis linking socioeconomic standing to AD, although the underlying cause for this association has not been unequivocally elucidated by prior research. A study by Wang et al. employed Mendelian randomization to delve into the potential influence of socioeconomic strata on AD and probed if elevated income exerted a protective effect against the disease’s onset [7]. From a health economics perspective, evaluations bifurcate into comparative analyses, assessing the cost–benefit ratio of varied therapeutic avenues, and cost-of-illness (COI) evaluations, which ascertain the economic strain of an ailment from a defined standpoint. Parkinson’s disease has been the subject of numerous COI studies across diverse global regions [8]. The protracted nature of PD, characterized by escalating disability and increasing dependence in activities of daily living (ADLs), imposes a substantial socioeconomic load. Advanced stages necessitate specialized institutional care, entailing significant resources and expenditures. Moreover, the familial impact of PD is profound with most ADL-dependent patients relying on home-based care provided by family members [9]. A subsequent exploration hypothesizes that there is a potential correlation between ALS risk and dietary habits, specifically the frequent intake of expensive, high-trophic level fish species known for elevated mercury content. This led to a detailed examination of the interplay between ALS, socioeconomic status, and mercury exposure via fish consumption [10]. Furthermore, the CDC’s NIOSH National Occupational Mortality Surveillance (NOMS) research discerned that professions linked to a superior socioeconomic tier, such as computer-related fields, engineering, legal practices, and business operations, manifested augmented ALS mortality rates after adjusting for demographic variables like age, gender, and ethnicity [11].

### 1.3. The Critical Need for New Biomarkers in the Context of Disease Diagnosis, Monitoring, and Treatment

The discernment of consistent biomarkers holds promise in advancing the early detection of neurodegenerative diseases, paving the way for the initiation of tailored therapeutic regimens. At present, the realm of epigenetics lacks robust and dependable biomarkers conducive to the diagnosis, categorization, or tracking of neurodegenerative disease progression [12]. In the context of available diagnostic modalities for neurodegenerative ailments, while pathological evaluations are held in high esteem across diverse afflictions, their applicability is limited in discerning neurodegenerative diseases during a patient’s lifetime. Alternatives like positron emission tomography (PET) scans or emergent biomarkers (encompassing genomics and proteomics) present potential breakthroughs and are being integrated into refined diagnostic parameters [13]. However, it is noteworthy that parameters such as DNA methylation levels, SIRT activity, and BDNF expression witness a marked decline in individuals diagnosed with dementia or Parkinson’s disease. Hence, the concurrent assessment of these epibiomarkers might enhance the diagnostic accuracy for neurodegenerative diseases. Given the reversibility of epigenetic alterations, gauging parameters like DNA methylation levels, SIRT activity, and BDNF expression could equip medical practitioners with insights to evaluate the efficacy of therapeutic interventions [14].

## 2. The Significance of Biomarkers in Neurodegenerative Diseases

### 2.1. What Are Biomarkers and Why Are They Important?

The concept of a “biomarker”, derived from the amalgamation of “biological” and “marker”, encompasses a broad range of medical signs. These signs provide objective evidence of a patient’s health condition and can be consistently and accurately quantified. This is distinct from medical symptoms, which are subjective sensations or complaints reported by the patient [15]. Biomarkers serve as pivotal tools in the methodical evolution of pharmaceuticals and medical apparatuses [16]. Yet, despite their immense significance, there exists a pronounced ambiguity surrounding their foundational definitions and the intricacies of their application in both research and clinical settings [17]. The spectrum of biomarkers ranges from elementary metrics like pulse and blood pressure to intricate laboratory assessments of blood and other biological specimens. Historically, medical signs have always been integral to clinical practice, with biomarkers representing the pinnacle of objective and quantifiable indicators that contemporary lab sciences can consistently measure. In the realm of drug innovation and broader biomedical investigations, biomarkers hold a transformative role. Deciphering the interplay between quantifiable biological mechanisms and clinical results is paramount for bolstering our repertoire of disease interventions and for a profound comprehension of standard physiological processes [18]. For biomarkers to be genuinely efficacious as replacements for clinically relevant endpoints, there is a prerequisite to thoroughly grasp the standard biological mechanisms, the alterations in disease conditions, and the impacts of varied interventions, be they drug-induced, device-based, or other [15]. The imperative for the prompt and precise identification of neurodegenerative conditions in clinical environments cannot be overstated. Beyond furnishing diagnostic and future insights, this need also encompasses the fine tuning of therapeutic approaches, ensuring apt care and support, and offering patients avenues to participate in clinical therapeutic studies [19].

### 2.2. Differentiating between Risk, Prodromal, Clinical, Wet, Dry Markers and Surrogate Endpoints

The methodology of risk assessment finds its application across diverse clinical spheres and for a variety of clinical outcomes. Regardless of the specific clinical domain or outcome in question, the foundational principles and techniques for evaluating risk markers and risk assessment remain consistent. Risk is typically gauged by counting the number of outcome incidents over a specified time span. This is traditionally encapsulated either via a survival curve or by denoting the fraction of incidents within a designated time frame, such as 30 days or a year [20]. As a result, there is often a strong interrelation among multiple biomarkers, complicating the process of pinpointing a singular prominent marker. Within the field of periodontology, the quest for risk biomarkers that can predict potential disease onset in individuals devoid of clinical symptoms is ongoing [21]. For Parkinson’s disease in its prodromal phase, while markers can facilitate diagnosis, it is imperative to understand four central characteristics of these markers, especially if they are to guide the selection of neuroprotective treatments. Among these, understanding the specificity or predictive accuracy of the marker is crucial, given the notable variances in specificity and positive predictive value (PPV) among different prodromal markers [22]. In this context, a “wet biomarker” is delineated as a prospective biomarker that can be objectively ascertained within a body fluid [23]. Biomarkers have been categorized into two main types: “dry” markers, which encompass imaging parameters, and “wet” markers, which refer to genetic and biochemical elements detectable in fluids such as blood, serum, urine, and tissue samples [24]. There are also surrogate markers (or surrogate endpoint), which are markers that are used as a distant relationship between an action and a clinical endpoint. An example of this would be the easy-to-understand relationship between smoking and lung cancer [25]. A surrogate endpoint of smoking would be death. Therefore, smoking is a surrogate marker of death via lung cancer. The utility of these endpoints would be of great value because it would clarify more easily the barrier between the general population and disease. It is a challenging task to pick surrogate endpoints and demonstrate their efficacy, because this action requires an extraordinary understanding of the disease’s pathophysiology. Several studies in the current literature have clarified the important yet difficult task to create these surrogate endpoints, and they have demonstrate the failure of this viewpoint in numerous studies, including neurodegenerative disease [26,27,28,29,30].

### 2.3. Overview of Their Roles in Early Diagnosis, Monitoring Disease Progression, and Evaluating Therapeutic Efficacy

At present, the categorization of most biomarkers hinges on the pathogenic processes they signify. For conditions like Alzheimer’s disease and frontotemporal lobar degeneration (FTLD) spectrum, the primary focus is on biomarkers indicative of pathology, such as those for amyloid-β (Aβ) and tau pathologies. These biomarkers are predominantly evaluated through CSF examinations, blood tests, and positron emission tomography scans [31]. In the preclinical stages of AD, while there are detectable biomarkers signaling brain alterations, clinical manifestations remain absent [32]. Conversely, in Parkinson’s disease (PD), the onset of classic motor symptoms is observed only after a significant proportion, over half, of neurons in the substantia nigra (SN) have already degenerated [33]. Consequently, pinpointing these conditions early is imperative for implementing strategies geared toward preventing neuronal loss. Over recent years, there has been a concerted effort by researchers to bolster the advancement of reliable biomarkers for neurodegenerative ailments. Despite these endeavors, results have often been inconsistent and not always meeting optimal standards. The trajectory of medical practice is increasingly leaning toward precision medicine, underscoring the pressing need to seamlessly incorporate disease-specific biomarkers in clinical routines and to engineer potent disease-altering treatments [31]. A double approach regarding neurodegenerative disease could be, firstly, neuroinflammation, which is a key factor that is both result and cause of neurodegeneration [34]. Secondly, in the last decade, research has pinpointed another key factor of neurodegeneration: cIMT (carotid intima media thickness). cIMT has been long debated as a surrogate endpoint of neurodegenerative disease; however, nowadays, it is a relevant influence in neurodegenerative disease [35,36,37].

### 2.4. Overview of Biomarkers in Huntington’s Disease, Multiple Sclerosis, Frontotemporal Dementia and Essential Tremor

#### 2.4.1. Huntington’s Disease

Increasing emphasis has been placed on the significance of white matter in the degenerative process [38], as widespread alterations can be detected over a decade prior to anticipated disease onset [39]. A comprehensive study amalgamated clinical and morphometric imaging data from 1082 participants, sourced from the IMAGE-HD, TRACK-HD, and PREDICT-HD studies, with longitudinal observations spanning 1–10 years. The findings from this research indicate that imaging might be a viable endpoint in clinical trials due to its potential heightened sensitivity [40].

Regarding the wet biomarkers, a study indicates that mutant HTT levels exhibit correlations with clinical scores both cross-sectionally and in relation to CSF tau and neurofilament light chain (NfL) [41], both being indicators of neuronal damage [42]. This suggests that mHTT is likely released from compromised or deteriorating neurons. Given the pivotal role of mHTT in HD pathogenesis, it emerges as a salient potential biomarker. Not only is it the pathogenic agent in itself, but in the context of Huntington-lowering, it stands as a crucial gauge of pharmacodynamics, signifying whether the therapeutic agent has effectively engaged its target and manifested the anticipated immediate biological effect [43] (See Table 1).

One study indicated that the accumulated data suggest a discernible segment of mHTT in the CSF is derived from striatal cells. These results advocate for the application of CSF mHTT as a PD biomarker in evaluating the engagement of therapeutic interventions tailored to decrease mHTT levels in the striatum [44].

A subsequent study explored the feasibility of utilizing noninvasive positron emission tomography (PET) for direct assessment of therapeutic efficacy and monitoring disease evolution in relation to mHTT. In this context, the novel radioligand [11C]CHDI-626 was characterized and examined longitudinally for mHTT PET imaging within the zQ175DN mouse model of HD. Notwithstanding its rapid metabolism and kinetics, the radioligand proved efficacious for mHTT PET imaging [45].

Liu et al.’s study furnishes initial evidence indicating that the early introduction of HTT-lowering treatment, prior to the manifestation of motor symptoms and striatal atrophy, can defer the onset and decelerate the progression of pathology and phenotype in a mouse model expressing full-length mHTT [46]. Concurrently, the research findings posit that the observed alteration in CBVa in premanifest zQ175 mice is a subsequent effect stemming from the influence of mHTT on neural activity/metabolism. Furthermore, the study suggests that a diminished rate of oxygen/nutrient delivery, attributed to a reduced cerebral blood volume and a decline in glucose transporter GLUT1 across a jeopardized neurovascular network during the manifest stage, may eventually instigate neuronal dysfunction and degeneration [46] (See Table 1).

#### 2.4.2. Multiple Sclerosis

In multiple sclerosis (MS), magnetic resonance imaging (MRI) elucidates the dimensions, quantity, chronology, and evolution of lesions within the central nervous system (CNS). Consequently, MRI is integral to the diagnostic process and therapeutic surveillance [47,48,49]. A study by Huang et al. demonstrates an up-regulation of MIP-1a and CXCL10 in the cerebrospinal fluid (CSF) of patients diagnosed with multiple sclerosis. Collectively, these cytokine biomarkers serve as a significant indicator of T cell activity, offering a measure that is both independent and complementary to the previously documented CXCL13, which is a chemokine targeting B lymphocytes [50]. To date, the singular cerebrospinal fluid (CSF) biomarker of clinical significance for MS is the presence of immunoglobulin G (IgG) oligoclonal bands (OCBs). These OCBs signify the intrathecal production of IgG, acting as a broader indicator of adaptive immunity activation within the CNS. It is pertinent to note that OCBs are not exclusive to MS; they have been identified in various inflammatory neurological disorders. Additionally, approximately 5% of MS instances do not exhibit CSF OCBs based on conventional assays [51,52,53,54,55] (See Table 2).

Blood-based serum neurofilament light chain (sNfL) is a potential and easily accessible prognostic and treatment response biomarker for patients diagnosed with multiple sclerosis. It is important to note that without the inclusion of supplementary clinical context, sNfL on its own does not suffice for diagnosing multiple sclerosis or distinguishing it from other neuroinflammatory conditions characterized by neuroaxonal damage and elevated sNfL levels, such as neuromyelitis optica spectrum disorders or myelin oligodendrocyte glycoprotein (MOG) encephalomyelitis [56,57,58,59].

#### 2.4.3. Frontotemporal Dementia

Over the past decade, neurofilament light chain (NfL) has garnered attention as a potential biomarker for FTLD due to its sensitivity in detecting neurodegeneration. Moreover, its levels demonstrate a correlation with the pace of clinical progression, providing prognostic insights. Recent scholarly investigations underscore the utility of NfL as a discriminative biomarker between bvFTD and primary psychiatric disorders, exhibiting areas under the curve ranging from 0.84 to 0.94 [60,61,62,63,64].

Progranulin (GRN) can be quantified in both blood and CSF, although the preponderance of research has been conducted on blood samples. Preliminary investigations reported remarkable sensitivity and specificity (both exceeding 95%) with a threshold of 61.5 ng/mL (ascertained in plasma using the Adipogen assay). However, subsequent research has proposed an elevated threshold of 71.0 ng/mL, boasting a sensitivity of 98.1% and specificity of 98.5%. It is posited that these levels are diminished from birth, as they appear to be low even when first assessed during late adolescence. Furthermore, these levels manifest consistent stability over extended periods, remaining relatively unaltered for up to four years as evidenced in one study [65,66,67] (See Table 3).

#### 2.4.4. Essential Tremor

In a forward-looking study that distinguished between sporadic-ET and hereditary-ET cases, the levels of uric acid were juxtaposed with those of controls. The results did not indicate significant deviations, thereby not affirming a neuroprotective function of uric acid in ET. Nonetheless, it is noteworthy that a correlation emerged between reduced uric acid levels and a later age of onset in sporadic cases, suggesting its potential significance as an indicator of neurodegeneration in such patients [68].

A study by Wang et al. introduced a methodologically sound consensus-based approach to scrutinize cerebellar involvement in ET, leveraging an augmented cohort for enhanced statistical power and taking into account the implications of MRI processing pipelines and statistical frameworks. This examination did not identify cerebellar involvement for advanced ET when synthesizing findings from three MRI biomarkers: voxel-based morphometry, cerebellar gray matter and white matter volumetry, and cerebellar lobular volumetry. The hypothesis was further assessed using ten prevalent statistical models based on biomarkers from Freesurfer, SUIT, and MAGeT. Notably, no cerebellar ROI derived from these three pipelines exhibited a consistent significant discrepancy [69] (See Table 4).

Another study performed by Yu et al. revealed that erythrocytic total and aggregated α-syn concentrations were significantly elevated in PD and ET patients in comparison to HCs. Notably, erythrocytic total α-syn levels were observed to be markedly higher in the ET cohort than in the PD group. Additionally, the ratios of erythrocytic aggregated to total α-syn levels in the ET group were discernibly reduced relative to those in the PD and HC groups. A significant correlation was also identified between erythrocytic aggregated α-syn levels and the disease duration in ET patients [70].

## 3. Alzheimer’s Disease (AD): The Quest for Novel Biomarkers

### 3.1. Current State of AD Biomarker Identification

Lecanemab (BAN2401), an IgG1 monoclonal antibody, is designed to target soluble aggregated forms of amyloid beta (Aβ), spanning oligomers, protofibrils, and insoluble fibrils. The BAN2401-G000-201 clinical trial, structured as a randomized double-blind study with a Bayesian design, evaluated three doses of lecanemab against a placebo in the early stages of Alzheimer’s disease, covering both mild cognitive impairment due to AD and mild AD dementia [31]. The primary evaluation criterion was the change from the outset at 12 months based on ADCOMS [71]. Essential secondary criteria encompassed changes in brain amyloid through PET Standard Uptake Value ratio (SUVr), ADCOMS, Clinical Dementia Rating-Sum-of-Boxes (CDR-SB), Alzheimer Disease Assessment Scale-Cognitive Subscale (ADAS-Cog14), CSF biomarkers, and total hippocampal volume as discerned by volumetric magnetic resonance imaging (vMRI). An additional focal point was the assessment of lecanemab’s efficacy in comparison to a placebo at 18 months based on ADCOMS, CDR-SB, and ADAS-Cog14 within specific clinical subgroups [72].

Given the lack of therapeutic solutions for AD, physical activity has emerged as a pivotal lifestyle determinant that might mitigate or delay the disease’s onset [73]. Delving into the impacts of exercise on systemic biomarkers linked to AD risk and correlating them with pivotal metabolomic shifts can propel preventive, monitoring, and therapeutic endeavors. A study evaluated systemic biomarkers, namely CTSB, BDNF, and klotho, and conducted a metabolomics analysis after a 26-week aerobic regimen [74]. In terms of CSF biomarkers, specific dietary patterns manifested varying effects on Aβ40 and Aβ42/40 ratios among different participant groups [75]. Another investigative endeavor probed into plasma biomarkers tied to neuroinflammation in relation to AD among a preclinical AD cohort. Only GFAP was found to be significantly elevated in the preclinical AD group compared to the healthy elderly [76]. See Figure 1.

Another study incorporated both subjective and objective cognitive performance metrics, in addition to parameters like sleep, stress, mood, and quality of life, facilitating a comprehensive evaluation of cognitive function and psychosocial well-being in relation to AD biomarker shifts [77]. An exploration into the relationship between certain plasma biomarkers and clinical efficacy endpoints underscored the predictive capacity of these markers in gauging cognitive decline [78]. The study also highlighted the potential utility of plasma biomarkers in monitoring lecanemab’s therapeutic effects and possibly individual patient responses. These insights are formative and will be delved into further in upcoming phase 3 lecanemab clinical trials [78]. Additionally, gantenerumab treatment showcased a dose-dependent impact on CSF biomarkers indicative of AD’s core pathological processes [79], including synaptic dysfunction [80]. An exploratory analysis from the TRAILBLAZER-ALZ study indicated that donanemab treatment modulates plasma levels of specific biomarkers relative to a placebo with these changes correlating with amyloid plaque shifts as identified by amyloid PET imaging [81].

### 3.2. Novel “Wet” and “Dry” Markers in the Horizon

The current understanding of biomarkers hinges on their capability to provide insights into the underlying mechanisms of AD pathology. Established CSF analytes have shown varying degrees of accuracy across different studies, emphasizing the urgency for innovative biomarkers that can enhance diagnostic precision. There is a growing demand for biomarkers that can elucidate additional aspects of AD pathogenesis, highlighting areas like neuroinflammation and early neuronal dysfunction preceding overt cell death [82]. An extended study, named ALFA+, seeks to conduct the in-depth phenotyping of a subset of participants from the ALFA parent cohort. The research will incorporate both wet (CSF, blood, urine) and imaging (MRI and PET) biomarkers for a holistic evaluation [83].

The REST protein has emerged as a potential novel biomarker for AD, albeit its exclusive detection in the central nervous system and in vitro models limits its application in translational research [84]. A trend in REST levels has been observed with declining levels corresponding to increasing clinical severity of the disease. Preliminary findings suggest certain biomarkers, such as NPTXR, might be indicative of AD progression [85]. The APOE ε4 allele stands out as a significant genetic determinant for AD susceptibility with carriers exhibiting distinct pathological traits, including a higher prevalence of amyloid plaques [86]. A comprehensive analysis of 30 brain-centric proteins as potential CSF biomarkers for AD progression revealed NPTXR as a promising candidate with levels decreasing commensurately as AD advances [87].

CSF samples, pivotal in these evaluations, were diligently processed and analyzed at the Leonard Wolfson Biomarker Laboratory, University College London. Standardized protocols were employed for the assessment of Aβ42, T-tau, and P-tau [88]. The VIVIAD trial, a phase 2b study, is evaluating varoglutamstat’s potential as a disease-altering treatment for AD with an emphasis on its correlation with both novel and traditional biomarkers [89]. Biomarker alterations serve as crucial outcome metrics in phase 2b AD trials. For instance, the SAPHIR trial observed a decline in YKL-40, a marker for AD-related neuroinflammation, following varoglutamstat administration [90]. As the field of biomarkers evolves, developing advanced nanotechnologies to monitor neuronal activity within networks, especially focusing on [Ca^2+^] changes in living organisms, will be a pivotal challenge [91].

### 3.3. How Emerging Technologies Are Aiding in the Identification of New Markers for AD

Neuroinflammatory markers, pivotal in understanding neurodegenerative processes, have been evaluated using the Luminex xMAP technology in significant cohort studies. However, these investigations have produced inconsistent results, which poses challenges, especially for academic entities and small biotechnology companies striving to develop treatments targeting neuroinflammation [92]. Although there have been substantial advancements in technology and methodologies for target identification and evaluation, the journey from recognizing promising targets to early drug discovery remains intricate and uncertain [93].

Metabolomics, with its diverse analytical platforms, offers powerful diagnostic tools and insights into disease mechanisms. These technologies have been employed in both animal and human studies, encompassing plasma and CSF evaluations. They have identified metabolic pathways that are disrupted in conditions like AD and MCI [94]. A notable discovery is the reduced plasma levels of desmosterol, a cholesterol precursor, in AD patients. This decrease correlates with cognitive changes, suggesting its potential as an AD diagnostic biomarker. Merging metabolomic signatures with other biomarkers could further enhance diagnostic specificity [95] (See Table 5).

DNA microarray techniques have also been employed to delve into neurobiology and neurodegeneration. Recent publications have emphasized the significance and appropriate utilization of this technology while exploring neurodegenerative mechanisms [96]. Innovative imaging technologies, such as PET, hold promise for enhancing early diagnostic precision in AD’s prodromal states, especially in patients with MCI, potentially fast tracking the evolution of disease-altering treatments [97]. The emergence of high-throughput DNA genotyping and sequencing has facilitated numerous genome-wide association studies (GWASs) in AD [98].

The arena of stem cell technology is witnessing rapid advancements. Many patients are opting to have their stem cells collected and reprogrammed. One advantage is the creation of cellular models representing “aged” cells, but there is caution to exercise, as reprogrammed cells may not perfectly replicate native neurons [91].

## 4. Parkinson’s Disease: Beyond Traditional Markers

### 4.1. Challenges in Early Diagnosis and Monitoring of PD

A family history showcasing a similar tremor pattern may point toward essential tremors, particularly given that this condition often exhibits an autosomal dominant inheritance pattern. Conversely, indicators such as a classic rest tremor, primarily unilateral tremor presentation, leg tremor, associated rigidity, and a response to levodopa are suggestive of Parkinson’s disease [99]. The diagnosis of idiopathic Parkinson’s disease is still largely clinical despite technological advancements in radiological assessments. Distinct clinical signs required for diagnosis include a distal resting tremor ranging between 3 and 6 Hz, rigidity, bradykinesia, and an asymmetrical onset [100]. Other hallmark signs encompass late-onset postural instability, olfactory deficits, and micrographia.

Machine learning (ML) has been harnessed by researchers aiming for early Parkinson’s disease diagnosis, utilizing motion data gathered from individuals’ upper limbs [101]. Experiments had participants, both those diagnosed with PD and healthy individuals, wear a device on their upper limbs while performing specific tasks [102]. To determine the optimal model for PD diagnosis, numerous experiments were conducted. The selected network topology comprised a single hidden layer with eight neurons. Tanh, Relu, and sigmoid functions were designated as activation functions for input, hidden, and output layers, respectively [102].

Early clinical diagnosis of PD is intricate, as overt differences in motor and cognitive features are elusive. Comprehensive understanding of clinical symptoms, pathological alterations, and neural dysfunction is imperative for a definitive disease diagnosis [103]. While many ML-based models have been proposed for ESPD diagnosis [104], the BNA neuromarker, derived from easily obtainable EEG data, stands out for its clinical utility and repeatability [105]. Alongside therapeutic interventions like gene therapy, neuroprotection, and pharmacology, the search for PD’s biological markers is relentless, aiming at early diagnosis [103].

Balance training, in particular, suffers from the lack of standardized approaches in monitoring training programs, making incomplete descriptions problematic [106]. During the trial, the influence of both study treatments, CBT and clinical monitoring, on depression in Parkinson’s patients remained uncertain. However, factors such as the chronic depression experienced by the sample, the progressive nature of PD, and the durable gains from CBT over 14 weeks suggest that the benefits of CBT might surpass mere placebo effects [107]. Digital biomarkers have shown potential in passive monitoring, indicating decreased mobility in PD participants relative to controls. These biomarkers could detect significant irregularities even when traditional exams did not, hinting at their heightened sensitivity, making them suitable for long-term clinical trials and treatment monitoring [108]. In scenarios where therapy response is subpar and alternative explanations are absent, more advanced methods like electronic compliance monitoring may prove beneficial [109].

### 4.2. Innovative “Wet” and “Dry” Biomarkers for PD

NTK stands as a previously identified biomarker panel, which was validated through a comprehensive, longitudinal study involving 2743 early AD patients. During this study, multiple CSF biomarkers exhibited notable alterations [110]. In relation to Parkinson’s disease, there has been a documented decline of 10–15% in CSF αSyn in comparison to healthy controls (HCs) [111]. This discovery was further confirmed using an independent methodology. However, this research stands as the inaugural longitudinal CSF study that focused on PD and HC using this specific biomarker panel. Notably, apart from αSyn, the study found no significant variations in other evaluated biomarkers [112] (See Table 6).

The T1w/T2w ratio within the midbrain is considered to embody a culmination of multiple PD-associated changes. These include modifications in neurons, dendrites, microglia, and iron content. Such data might produce a pronounced contrast that could be more effective than alternative MRI sequences in detecting PD-associated pathology. This ratio could potentially serve as an early detection biomarker for PD. To further this hypothesis, a subsequent MRI–pathology correlation study is recommended [113].

There have been indications that platelet CoQ10 redox ratios are considerably reduced in PD patients [114]. However, this test has not transitioned into clinical applications yet. The identification of a peripheral biomarker that can recognize decreased coenzyme Q10 activity may expedite research and improve clinical outcomes concerning PD [115].

Sargramostim, when administered in low doses, has shown the potential to modify immune functions, influence T cell phenotypes, and amplify treatment-induced biomarker levels. These changes have been associated with improved MDS–UPDRS Part III scores. The treatment also amplified Treg-mediated immunosuppressive functions, which remained consistent throughout the study. It is noteworthy that Tregs from PD patients previously exhibited a hindered ability to suppress Teff proliferation, which was linked to heightened disease severity [116].

Additional biomarkers like α-synuclein, neurofilament light chain, tau, phospho-tau, and beta-amyloid were assessed as potential exploratory endpoints over a 4-week treatment period. However, this duration might have been insufficient to detect significant clinical changes in these parameters. It is important to mention that there are not any validated biomarkers for PD presently. Future research endeavors might investigate the influence of venglustat on biomarkers and the progression of the disease over extended treatment periods [117] (See Table 6).

The effects of nilotinib on CSF biomarkers suggest that reducing oligomeric α-synuclein and p-tau could enhance dopamine metabolism in PD patients [118]. The data from both clinical and biomarker perspectives indicate that pioglitazone may not be a promising neuroprotective agent for PD. An intriguing point is that even though an epidemiological study pinpointed a reduced PD risk among individuals exposed to glitazone drugs, this association was not validated in a subanalysis that was specific to pioglitazone [119]. In this study, DaT-SPECT was employed as a PD enrichment biomarker, unveiling a SWEDD incident rate (3.8%). This rate was considerably lower than what is typically observed in multiple large multicenter studies with analogous PD populations [120,121].

**Table 6 ijms-24-16119-t006:** The wide variety of biomarkers used in Parkinson Disease and their clinical association.

References	Biomarker(s) or Indicator	Association/Significance
[99]	Classic rest tremor, Unilateral tremor presentation, Leg tremor, Associated rigidity, Response to levodopa	Indicators suggestive of Parkinson’s disease
[100]	Distal resting tremor (3 to 6 Hz), Bradykinesia, Asymmetrical onset, Late-onset postural instability, Olfactory deficits, Micrographia	Clinical signs required for Parkinson’s disease diagnosis
[105]	BNA neuromarker (from EEG data)	Stands out for its clinical utility and repeatability in ESPD diagnosis
[108]	Digital biomarkers (mobility)	Potential in passive monitoring indicative of decreased mobility in PD participants
[111]	CSF αSyn decline	Documented decline in Parkinson’s patients compared to healthy controls
[113]	T1w/T2w ratio within the midbrain	Could serve as an early detection biomarker for PD due to various PD-associated changes
[114]	Platelet CoQ10 redox ratios	Indicative of reduced platelet CoQ10 redox in PD patients
[116]	Treatment-induced biomarker levels (Sargramostim)	Association with improved MDS-UPDRS Part III scores and modified immune functions
[117]	α-Synuclein, neurofilament light chain, tau, etc.	Assessed as potential exploratory endpoints, but duration was potentially insufficient for significant changes
[118]	Oligomeric α-synuclein and p-tau (effects of nilotinib)	Suggest that reducing these could enhance dopamine metabolism in PD patients
[119]	DaT-SPECT	Employed as a PD enrichment biomarker

### 4.3. Potential for Tailored Therapies and Improved Diagnostic Accuracy

Currently, Alzheimer’s disease treatment employs only three AChE inhibitors: donepezil, rivastigmine, and galantamine. These medications serve primarily to offer symptomatic relief and are predominantly prescribed for mild to moderate dementia cases [122]. Art therapy has demonstrated notable benefits for patients, such as enhanced visual exploration patterns that begin to align with those of a control group. This suggests that art-centric visual training can foster the adoption of efficient visual exploration techniques [123]. To elaborate, art therapy has been shown to yield significant enhancements in visuospatial abilities, visual exploration strategies, and motor functions in PD patients with mild to moderate impairment. These improvements coincide with functional connectivity (FC) changes, pointing to a functional reorganization within primary and associative visual networks. This indicates that art therapy may serve as a valuable supplementary treatment to existing pharmacological interventions [123].

The core objective of a specific study was to ascertain if a customized tai chi program could bolster postural stability in Parkinson’s disease patients [124]. The results revealed that practicing tai chi twice weekly for 24 weeks, in comparison to resistance training or stretching programs, effectively enhanced postural stability and other functional aspects in patients with mild-to-moderate Parkinson’s disease. Moreover, tai chi training led to a marked reduction in fall incidents compared to the stretching routine. These positive outcomes persisted three months post-intervention, aligning with prior studies focused on individuals aged 70 and above [124]. Given the chronic and progressive nature of PD, it is recommended that the visual feedback VR technique be adopted as a long-term treatment strategy, complementing physical therapy, to sustain gait and postural performance in PD patients [125].

A study used a classification method which was validated using LOOCV and achieved an impressive classification accuracy of 93.62%. Most of the altered functional connections that exhibited high discriminative power were predominantly found within or across specific networks and the cerebellum [126]. Some studies managed to obtain a high classification accuracy of 94.4%, but the employed imaging method was invasive, making it unsuitable for routine diagnostics [127]. In contrast, certain noninvasive techniques have attained commendable classification accuracy using multi-type feature combinations. However, none have reached the high accuracy levels of this classification results [128]. The classification model, incorporating the basic SVM model and FG III, surpassed other ensemble classification models in performance. The final ensemble model was assessed using independent test data, achieving a 75.8% accuracy in distinguishing between early-stage PD and ET [129]. While the early-stage PD and ET classification model showcased good feasibility and potential, it was not exceptional [129].

The reliability of assessments using wearable sensors is influenced by factors such as sensor positioning, sensor-to-segment alignment, and frequently, the total number of sensors. This often results in increased costs and obtrusiveness [130]. In contrast, devices like the Wii Balance Board (WBB) and the Kinect for Windows v2 provide potential solutions for human motion tracking, circumventing the challenges posed by wearable sensors [131].

## 5. Amyotrophic Lateral Sclerosis (ALS): The Frontier of Biomarker Research

### 5.1. Overview of the Unique Challenges Posed by ALS

While the fundamental definition of ALS appears clear-cut, emerging insights suggest that ALS is not a singular disease but encompasses a diverse array of conditions with shared clinical characteristics [132]. People diagnosed with ALS face unique challenges compared to other patient groups where expressive disclosure has been employed as a therapeutic strategy, such as those with cancer, rheumatoid arthritis, or asthma. This uniqueness stems from the rapid progression of ALS, leading to paralysis, loss of independence, communication barriers, and the inevitable fatal prognosis. Given the swift and dynamic nature of ALS, the physical and emotional hurdles faced by patients may evolve significantly within a span of six months post-intervention. As such, emotional expression interventions tailored for ALS and similar rapidly progressing diseases might benefit from periodic ‘booster’ sessions. These sessions can address the evolving challenges and emotional shifts that patients encounter as the disease progresses [133]. See Figure 2.

Caregivers attending to ALS or progressive muscular atrophy (PMA) patients navigate numerous challenges as they witness the relentless progression and fatal trajectory of the disease. They grapple with the physical decline of the patient and potential cognitive and behavioral changes, escalating the caregiver’s responsibilities and emotional strain [134]. Prolonged clinical research has identified systemic metabolic irregularities in ALS patients. While some discrepancies remain, a significant portion of studies highlight disturbances in functional metrics, like diminished glucose tolerance, insulin resistance, and abnormal fatty acid utilization [135]. However, the evident shortcomings of existing models in replicating ALS-specific conditions and associated pathologies cast doubt on their applicability for ALS research. The absence of accurate TDP-43 and FUS disease models represents a significant hurdle in ALS research. There is an urgent need for alternative models that can faithfully capture all dimensions of the disease [136].

### 5.2. Newly Identified Biomarkers and Their Potential Implications

The strategy of focusing on easily obtainable biofluids and evaluating markers directly linked to ALS pathogenesis stands as a cornerstone for the effective development of biomarkers. The occurrence of ferroptosis in motor neurons is increasingly acknowledged as a vital aspect of ALS with markers like lipid and iron accumulation signaling this specific type of programmed cell death [137,138]. Neurofilament light chain (NfL) and phosphorylated heavy chain (pNfH) are renowned indicators of neural integrity specific to ALS [139]. Within the Mitotarget/TRO19622 study, a cohort comprising 512 ALS patients from 15 European centers engaged in a phase III trial of olesoxime, and these biomarkers were assessed [140]. Notably, higher baseline levels of NfL, 4-HNE, 8-oxo-dG, and FT were linked with a steeper decline in ALSFRS-r during an 18-month monitoring period. Intriguingly, alterations in these markers outpaced functional deterioration with discernible differences between rapid and slow disease progressors observable at the 6-month mark [141].

Analyzing both the MCP-1 and FOXP3 mRNA, distinct effects were observed within the PP population. Among other findings, a significant change over time was solely detected in the PP population for actin-NT with no other notable effects identified for other biomarkers in the studied populations. To detect a 44% decrease in the progression rate of PPIA with 80% power, a total of 142 patients was deemed necessary. Likewise, the study aimed to discern a 43% reduction in ALSFRS-R progression over 24 weeks and a 25% absolute decrease in patients becoming non-self-sufficient at the 24-week mark [142].

Incepted in 2007, the Pre-Symptomatic Familial ALS (Pre-fALS) study is a longitudinal examination of unaffected individuals with a heightened genetic susceptibility to ALS. Its objectives encompass characterizing the pre-symptomatic disease phase, pinpointing biomarkers indicative of the imminent clinical manifestation, and collating essential data to pave the way for early intervention or preventive trials [143]. The ATLAS initiative holds promise in unearthing early markers of disease activity that extend beyond NfL. The periodic collection of CSF and urine/blood samples will facilitate the discovery of other potential fluid markers indicative of disease activity. Additionally, thorough electromyography (EMG) could provide insights into the temporal relationship between NfL elevation and the appearance of EMG anomalies, shedding light on the comparative sensitivity of these biomarkers [144]. Previous research by Keizman et al. pinpointed a notable correlation between clinical disability in ALS patients and inflammatory biomarkers, including CRP. This emphasizes the critical role of inflammation in ALS, underscoring CRP as a readily obtainable biomarker from blood samples irrespective of the patient’s clinical status. Elevated CRP levels have also been detected in the cerebrospinal fluid of ALS patients, accentuating the importance of neuroinflammation in the disease progression [145].

### 5.3. Technological and Methodological Advancements in ALS Biomarker Discovery

The Mitotarget/TRO19622 study was a phase III trial focused on olesoxime, which was carried out as a negative, randomized, double-blinded, and placebo-controlled trial. This trial incorporated 512 ALS patients drawn from 15 European centers [140]. This research strictly adhered to the guidelines and regulations set by both French and European authorities. The objective of this study was to evaluate the influence of RNS60 treatment on potential markers indicative of inflammation and neurodegeneration in the peripheral blood of ALS patients. The markers under scrutiny included MCP-1, PPIA, actin-NT, 3-NT, IL-17, NfL, and Tregs, which were identified through FOXP3 and CD25 mRNA [142] (See Table 7).

Part A of the study served as the natural history run-in phase. Throughout this period, participants underwent monthly monitoring to identify changes in their plasma NfL levels or the onset of clinically evident ALS. The design of Part A, which prioritized feedback from the ALS community, aimed to ensure minimal inconvenience for the participants. As such, most assessments, such as monthly blood draws for NfL monitoring, could be conducted within the confines of participants’ homes. Part B of the study was a randomized phase, which was double-blind and placebo-controlled. Here, pre-symptomatic participants exhibiting elevated NfL levels were randomized to either receive tofersen or a placebo. This randomization process was dynamic, factoring in aspects like SOD1 variant type, the last recorded plasma NfL level before randomization, and age [144].

To delve into the prognostic potential of CRP, serum levels were gauged at the inception of the study. These levels were then correlated with various clinical demographics of ALS patients, such as age at the time of diagnosis, gender, disease duration by the time of evaluation, onset site, ALSFRS-R total score, body mass index, smoking habits, and overall survival [145]. Following intrathecal infusions, participants’ cells were chased using CSF drawn prior to the transplantation process. After this process, participants were advised to maintain a specific position, the Trendelenburg position, for a duration of up to two hours. Throughout the study, participants, trial investigators, and personnel from the sponsor remained blind to treatment allocations. These allocations were assigned at the cell culture manufacturing facility once the clinical site informed them of participant eligibility [146].

In another study, patients were given either a placebo or increasing doses of NP001. The main endpoints for monitoring were safety, shifts in clinical status, and the reactions of blood monocyte immune activation markers CD16 and HLA-DR to NP001. These values were sourced from an independent flow cytometry laboratory at UCSF, which employed validated procedures for determinations. The statistical analysis for these values was conducted independently for CD16, while Neuraltus scientists managed the analysis for HLA-DR values [147] (See Table 7).

**Table 7 ijms-24-16119-t007:** New avenues in biomarker development of Amyotrophic Lateral Sclerosis.

References	Biomarker(s) or Indicator	Association/Significance
[94,95]	Ferroptosis markers (lipid and iron accumulation)	Linked to ALS-associated programmed cell death
[139]	Neurofilament light chain (NfL)	Indicator of neural integrity specific to ALS
[139]	Phosphorylated heavy chain (pNfH)	Indicator of neural integrity specific to ALS
[141]	NfL	Higher baseline levels linked with steeper decline in ALSFRS-r
[141]	4-HNE	Linked with steeper decline in ALSFRS-r
[141]	8-oxo-dG	Linked with steeper decline in ALSFRS-r
[141]	FT	Linked with steeper decline in ALSFRS-r
[142]	MCP-1	Observed distinct effects in PP population
[142]	FOXP3 mRNA	Observed distinct effects in PP population
[142]	Actin-NT	Significant change over time detected in the PP population
[142]	PPIA	Associated with progression rate
[145]	CRP (inflammation marker)	Correlated with clinical disability; role in inflammation
[142]	MCP-1	Indicator of inflammation and neurodegeneration
[142]	PPIA	Indicator of inflammation and neurodegeneration
[142]	Actin-NT	Indicator of inflammation and neurodegeneration
[142]	3-NT	Indicator of inflammation and neurodegeneration
[142]	IL-17	Indicator of inflammation and neurodegeneration
[142]	NfL	Indicator of inflammation and neurodegeneration
[142]	Tregs (identified through FOXP3 and CD25 mRNA)	Indicator of inflammation and neurodegeneration
[144]	Plasma NfL levels	Used for monitoring onset of clinically evident ALS
[147]	Blood monocyte immune activation markers CD16	Monitored for reactions to NP001
[147]	HLA-DR	Monitored for reactions to NP001

## 6. Emerging Susceptibility and Trait Markers for Neurodegenerative Diseases

### 6.1. Introducing the Importance of Susceptibility and Trait Markers

The influence of cumulative lead exposure on cognitive functions may be mediated by the APOE genotype. Specifically, the E4 allele of the APOE gene is a recognized risk factor for Alzheimer’s disease. Research has shown that individuals carrying at least one E4 allele, as opposed to those without the E4 allele, experience a more pronounced negative effect of bone lead on their neurobehavioral test scores, especially if they have been occupationally exposed to lead [148].

Certain studies indicate that the susceptibility to Parkinson’s disease from pesticide exposure might be influenced by alterations in genes responsible for detoxifying enzymes. For instance, a recent investigation revealed that changes in neuronal aldehyde dehydrogenase enzymes correlate with an elevated risk of developing PD [149].

Genetic susceptibility testing, which is currently witnessing rapid advancements, presents risk information that can be described as a “moving target.” The field of genetic testing is evolving quickly both in the variety of tests available and our growing comprehension of the intricate relationships between genes, environment, and behavior [150].

Quantitative Susceptibility Mapping (QSM) studies generally indicate heightened susceptibility, hinting at increased iron content, in brain regions linked to the pathophysiology of several neurodegenerative diseases. For instance, the substantia nigra in PD, the basal ganglia in Huntington’s disease (HD), the amygdala and caudate nucleus (CN) in AD, the motor cortex in amyotrophic lateral sclerosis, and the cerebellar dentate nucleus (DN) in Friedreich’s ataxia (FRDA) all show these changes [151].

Several studies have documented persistent sleep changes, such as shortened Rapid Eye Movement (REM) latencies, even during periods of remission from depression. Longitudinal research has consistently observed stable REM latencies, which suggests potential trait markers for some of these sleep alterations. This notion of a trait marker is further bolstered by the discovery of similar REM sleep changes in individuals with a pronounced family history of depression even if they were asymptomatic at the study’s time [152,153].

The parkin gene is expansive, spanning over 1.5 Mb with approximately 12 exons. It is located on chromosome 6q25.2-27. A specific mutation in this gene, specifically a homozygous exon depletion, was initially identified as a trait responsible for early-onset autosomal recessive Parkinson’s in a Japanese family [154].

### 6.2. Exploration of New Markers Identified through Genetic, Epidemiologic, and Epigenetic Studies

Biomarker data from the present research offer valuable insights into the disease’s mechanism of action, targeting several pathways, including neuroprotection, neuroinflammation, and neurodegeneration. Notably, all participants treated with MSC-NTF exhibited significant, consistent, and lasting changes in numerous neuroinflammatory and neurodegenerative biomarkers such as MCP-1 and NfL. These findings align with prior trials [146] and underscore the potential of a treatment associated with slowing disease progression [155].

In initial experiments with two cell lines, the accumulation of TDP43 fragments was diminished, and TDP-43 nuclear localization was reinstated when mTOR was inhibited by Rapamycin [156]. Furthermore, in both mouse and human stem cell-derived neurons and astrocytes containing mutant TDP43, enhancing autophagy led to improved TDP43 clearance and localization, emphasizing that autophagy induction counteracts neurodegeneration via TDP43 clearance [157].

The single nucleotide polymorphism (SNP) rs75932628-T has been associated with genetically higher sTREM2 levels in CSF and an increased risk of Alzheimer’s disease onset. Comparable results were observed for sTREM2 in CSF with a notable relative change from the baseline in the high exercise group versus the control group [158].

Evidence suggests a causal relationship between the LC and disease-modifying processes. Both genetic and neurotoxin-induced LC lesions exacerbate neuropathology and cognitive impairments in mouse models of AD, highlighting the LC’s pivotal role in regulating neuroinflammation [159]. Moreover, advanced techniques like DREADD chemogenetics and traditional pharmacological enhancement of NE neurotransmission can reverse AD’s pathophysiological features, boost microglial phagocytosis, and improve cognitive functions [160].

The detected levels correspond with those documented in extensive metabolizers. Anticipations for such outcomes were based on a pre-screening process designed to exclude potential carriers of CYP2D6 genetic variants, which are present in approximately 10% of the general population and are known to decelerate atomoxetine metabolism [118]. Nevertheless, it is essential to consider that CSF levels may be influenced by the permeability of the blood–brain barrier, factors correlated with aging, AD [119], and LC degeneration [120].

In the adopted reference-free methodology, all quantitative data post-assembly are disregarded. Under optimal circumstances, sequences deriving from identical genetic sources should culminate in a unique contig per sample. Within the framework of the BusyBee methodology, such a contig would emerge as a singular point, minimally impacting the comprehensive density distribution. The conspicuous signal emanating from the high-density cluster within the PD + RS group suggests the presence of multiple contigs. These contigs are sufficiently distinct to resist merging during assembly, yet they exhibit qualitative properties suggesting an association with Rhodococcus [121].

### 6.3. The Future Potential of These Markers in Personalized Medicine

Recent research has unveiled potential biomarkers that could prove instrumental in monitoring the progression of Parkinson’s disease. These findings also open up novel avenues for deeper investigation into the underlying mechanisms of PD. A study set out to validate these preliminary findings with additional sample sets. Furthermore, the study aimed to explore if the identified compounds that seem to predict the progression of PD can also distinguish between PD patients, healthy individuals, and those diagnosed with other neurodegenerative diseases [161]. Despite the consistent epidemiological associations between elevated urate levels and a decreased risk and progression rate of PD, the trial’s outcomes do not advocate for a protective role of urate [162]. Adding to the complexity, recent Mendelian randomization research challenges the protective nature of high urate levels against PD [163], while another study indicates its potential protective effect in slowing the progression of established PD [164]. The focus of the ATLAS study (NCT04856982) is to ascertain the effects of early administration of tofersen in individuals who are pre-symptomatic carriers of certain SOD1 mutations, which are known for their association with aggressive disease progression and increased plasma NfL levels. Given the potential benefits of early intervention in ALS, insights from ATLAS, combined with data from the VALOR study and its subsequent open-label extension, aim to provide clarity on the ideal timeframe for initiating treatment in cases of SOD1-ALS [144].

## 7. Diagnostic and State Markers: Revolutionizing Early Detection and Monitoring

### 7.1. Delving into Novel Diagnostic Markers for AD, PD, and ALS

There is growing emphasis on discovering new biomarkers that can bridge the gap between psychological risk factors and Alzheimer’s disease to foster a deeper understanding of the illness. Recent studies have highlighted the dysregulation of REST in depression, which is a psychological disorder linked with stress that elevates the risk for AD [165]. Given REST’s role in stress responses, it might serve as a pivotal biological link between psychological risks and AD. In exploration of the relationship between REST and previously pinpointed plasma protein markers associated with mild cognitive impairment (MCI) transitioning to AD and cortical atrophy [166], significant correlations were identified with BDNF, RANTES, PAI-1, and NSE. These correlations were independently validated in the Intervention cohort. BDNF, akin to REST, is believed to play a neuroprotective role under pathological conditions [167]. A cutting-edge PET method employing 11C-labeled AA was implemented, granting the first-ever visualization of in vivo dopaminergic neurotransmission in a resting state. Echoing findings from animal research [168], we observed significant increases in the incorporation coefficient K* for AA when exposed to apomorphine across several brain regions. This is believed to reflect neuronal signaling events associated with activated D2 receptors connected to cPLA2 [169]. NP001 is a specialized, pH-balanced stabilized sodium chlorite variant and presents a groundbreaking effector molecule that introduces a fresh drug category targeting inflammatory macrophages and modulating their function in vitro and in vivo [170]. Chlorite’s anti-inflammatory influence in macrophages is attributed to the elevated intracellular presence of taurine chloramine, which is known to suppress NF-κB triggered inflammatory pathways [171]. Prior clinical investigations with an alternate chlorite form have showcased its ability to counter inflammation and reset systemic macrophages to their natural wound-healing phagocytic state [172]. Contemporary studies indicate a direct correlation between the progression of the G93A strain of ALS mice and the infiltration of inflammatory monocytes into the spinal cord [147]. In this research, 30 brain-centric proteins were assessed as potential CSF biomarkers indicative of AD severity using multiplex mass spectrometry-based quantification. NPTXR emerged as a prime candidate for tracking disease progression. Intriguingly, two prior studies also flagged NPTXR as a promising progression biomarker for AD. As AD intensifies, CSF NPTXR levels proportionally decrease. This observation requires further validation in an expanded cohort observed longitudinally. It is hypothesized that NPTXR could be a pivotal CSF biomarker for gauging the effectiveness of emerging AD therapies [87].

### 7.2. Implications for Improved Diagnostic Accuracy

Studies of this domain unveil an intriguing observation: among older adults at a heightened risk of dementia, an 8-week stress reduction regimen led to a notable surge in REST levels, positioning REST as a potential adjustable target. The intricate role of REST in managing cortisol levels, primarily through the modulation of the CYP11B1 gene, [173] offers insight into the possibility that the intervention may have influenced cortisol concentrations, subsequently impacting REST levels. In our study, the selection criteria for the participant inclusion across the two cohorts either strictly involved individuals devoid of psychiatric ailments (like depression, anxiety, ANM) or those diagnosed with one (intervention cohort). Given this setup, the research could not draw a direct comparison of REST levels between mentally healthy seniors and those grappling with depression or anxiety. Consequently, a direct exploration between cognitive debt and this newfound biological indicator remains pending [84]. The innovative use of [1-11C]arachidonate PET in assessing healthy human participants showcased tangible impacts on the regional brain AA integration and rCBF following a pharmacological nudge with apomorphine, which is a D1/D2 receptor stimulant. The findings underscore the potential of this approach in capturing real-time signal transduction events tied to dopaminergic neurotransmission in a living brain. This paves the way for subsequent explorations into the efficacy of this technique in evaluating disruptions in cerebral dopaminergic functionality in disorders such as Parkinson’s disease and schizophrenia [174].

### 7.3. The Promise of These Markers in Disease Monitoring and Evaluating Drug Efficacy

Spatially normalized images were used to conduct the T1w/T2w ratio comparisons both for VBA and ROI-centric studies. A potential concern might be that these results could have been influenced by volumetric or morphometric data. However, this seems improbable, considering that the atlas-based segmentation approach (essentially the inverse of normalization) did not reveal any significant volume discrepancies between the PD patient group and the control group [113]. Building upon these insights, future endeavors might consider a phase II study that utilizes a seemingly immune-regulatory dosage of NP001 chlorite (2 mg/kg). This could be juxtaposed against a minimal effective dose (1 mg/kg) and a placebo, with the study span extended to discern if modulating inflammation influences the pace of ALS disease progression. Thus, research has laid the groundwork for deploying specific NP001 dosages targeting inflammation markers in ALS patients. This aims to explore the hypothesis that inflammation might play a pivotal role in the onset and development of ALS [147].

## 8. Progression Markers: Tracking Disease Evolution

### 8.1. Importance of Progression Markers in Neurodegenerative Diseases

Findings indicate that the diversity observed in Parkinson’s disease, especially concerning different ages of onset, might manifest through distinct deviations in both imaging and non-imaging biomarkers. When planning future clinical trials aiming to assess neuroprotective medications, it is crucial to factor in this biomarker variability associated with different PD onset ages. Opting for participants with a consistent age of onset could mitigate this variability, enhancing statistical power even with fewer participants. Continuous monitoring of the PPMI cohort will further elucidate the influence of onset age on the progression of PD and its potential interaction with these biomarkers [175]. The Parkinson Progression Markers Initiative (PPMI) operates as a global, multicenter cohort study, spanning 21 US and 12 international locations. This study focuses on patients newly diagnosed with PD who have not received any treatment at the point of enrollment, and it also includes healthy controls [176]. When it comes to understanding UPSIT scores, especially among individuals prone to or diagnosed with neurological conditions like PD, it is pivotal to rely on refreshed normative data. These data should ideally stem from a sizable sample that mirrors the demographic profile of PD patients. The UPSIT, given its adaptability for mail distribution and at-home self-administration, is perfectly aligned for large-scale investigations. It is worth noting that UPSIT has been employed in comprehensive studies such as Parkinson Associated Risk Syndrome (PARS) and the aforementioned PPMI. Our current research was tailored to offer normative data for UPSIT, segmented by age and gender, drawing insights from percentiles derived from the extensive, forward-looking cohorts of both PARS and PPMI [177].

### 8.2. Newly Identified Markers and Their Potential Role in Understanding Disease Trajectory

The observed lack of a significant statistical variation between the individuals undergoing DRT and those not undergoing DRT in terms of incident ICD symptoms may be attributed to multiple factors. The sample size being a limited one, the ability of distinguishment between the impact of dopamine agonists and other DRTs was diminished. Additionally, the potential variances in DRT and DAT availability among the four primary ICDs and behaviors such as punding, hobbyism, and walkabout were not thoroughly evaluated. The QUIP, which was primarily developed as a high-sensitivity (94%) screening tool but with a reduced specificity (72%) [178], might have resulted in certain participants displaying ICD symptoms that were either false positives or clinically non-pertinent [179]. Moreover, the study analyzed the expression levels of established apoptotic markers. These markers encompass pro-caspase 3, the p17 subunit of active caspase 3, cleaved PARP, and the anti-apoptotic protein BCL2. Notably, pro-caspase 3 is activated to form caspase 3 at the onset of the apoptotic process, leading to the proteolytic cleavage of the DNA repair enzyme, PARP, producing an 89 kDa apoptosis-specific PARP fragment [180]. In phase 2A of the study, the plasma concentrations of wr-CRP as potential biomarkers were quantified, and for phase 2B, hs-CRP plasma values were incorporated as a part of the enrollment criteria. To amalgamate CRP data from phase 2A and 2B for comprehensive analysis, a calibration equation derived from Ziv-Baran et al. [181] was employed to calibrate the phase 2A wr-CRP values. Baseline clinical and demographic data were analyzed according to their respective treatment groups [182]. The concluding evaluation of NP001 efficacy revealed a notably elevated percentage of non-progressors in the NP001 treatment group over a 6-month duration, in comparison to the placebo group. In patients displaying clinically relevant plasma CRP concentrations exceeding 3 mg/L, a 10:1 response favorability was observed for NP001-treated individuals versus placebo controls. Aligning with NP001′s anti-inflammatory properties, individuals displaying higher inflammation levels, as demarcated by blood CRP concentrations, exhibited a greater likelihood of benefiting from the treatment.

### 8.3. The Promise for Better Disease Management and Tailored Interventions

A reduction in DAT availability, especially a continuous decline over a period, might serve as an indicator for the likelihood of forthcoming ICD manifestations in early-stage PD patients post-initiation of DRT. Both neurobiological determinants and clinical attributes act as predisposing elements for the emergence of ICD symptoms in the context of DRT administration. Such understanding will aid in mitigating patient risks and devising innovative treatment strategies [179]. The potential influence of non-response bias merits attention. In the PARS study, 53% of qualifying participants submitted a completed UPSIT. Compared to non-participants, those who responded tended to be younger, female, Caucasian, have a familial history of PD, and did not indicate a diminished olfaction [183]. In the PPMI study, around 60% of the eligible cohort returned an UPSIT, but direct comparisons between participants and non-participants were not feasible. Regarding this specific study, interpretations concerning the prevalence of smoking in this amalgamated group are constrained as the data are exclusive to PARS participants. Such a limitation impedes the capacity to examine if smoking, linked with elevated olfactory dysfunction risk but reduced PD risk [184], might be a significant factor influencing the findings. An intriguing avenue for subsequent research would be an in-depth exploration of the interrelationships among smoking habits, olfactory function, and PD susceptibility [177]. Riluzole’s administration was largely well-received, with side effects being comparable to placebo. Riluzole has been a staple in ALS treatment for numerous years. Yet, to achieve a holistic assessment of riluzole’s safety and effectiveness within the Alzheimer’s demographic, extensive and prolonged studies are imperative before its administration to Alzheimer’s patients outside controlled clinical environments [185]. To sum up recent findings, riluzole-administered Alzheimer’s patients exhibited a more gradual decline in cerebral glucose metabolism compared to their placebo counterparts across various Alzheimer’s-relevant brain sectors. This decline was in correlation with their cognitive functionality. Such observations bolster the necessity for subsequent extensive clinical studies to further appraise riluzole’s potential as a prospective medicinal treatment for Alzheimer’s [186].

## 9. Innovative Techniques and Technologies in Biomarker Discovery

### 9.1. Highlighting the Role of -Omics, Histologic Assessments, and Imaging in Biomarker Identification

While numerous clinico-pathological and molecular biomarkers have been assessed for their potential benefits to FTD/TPI, their translation to clinical practice remains elusive [187]. Studies indicate that the KRAS mutational assessment, a globally recognized standard-of-care test, can discern patients with KRASG12 mutant mCRC who are less likely to derive benefits from FTD/TPI treatment. This identification aids in circumventing unnecessary patient side effects and optimizing healthcare resources. Consequently, this study presents the inaugural evidence of a genomics-driven precision approach for chemotherapy in mCRC, holding significant promise to enhance patient selection criteria for FTD/TPI therapeutic interventions [188].

### 9.2. The Synergy between Technology and Biomarker Discovery

Resveratrol appears to play a potential role in preserving the blood–brain barrier integrity primarily through the mitigation of MMP9 levels. Furthermore, resveratrol may stimulate adaptive immune mechanisms, potentially bolstering the brain’s resilience against amyloid accumulation. The compound’s potential to decelerate cognitive regression in Alzheimer’s disease may be attributed to a synchronized immune response, both peripheral and central, that could potentially halt neuronal apoptosis. Summarizing, the preliminary observations from the investigation underscore the need for a more extensive study to validate the supposition that resveratrol can fortify a compromised BBB, subsequently leading to cognitive and functional enhancements in a broader AD patient cohort [189].

## 10. Conclusions and Future Outlook

### 10.1. A Reflection on the Advancements Made and the Challenges Ahead

Exenatide, a therapeutic agent classified as a glucagon-like peptide 1 agonist and traditionally employed for type 2 diabetes, has recently demonstrated potential positive impacts on motor functionalities in a controlled trial involving Parkinson’s disease patients. Prevailing research posits that dysfunctional brain insulin and protein kinase B (Akt) signaling might be implicated in PD development. Nonetheless, comprehensively assessing the degree of drug interaction with these potential mechanisms in a live setting presents considerable difficulties [190]. Neurodegeneration, an intrinsic aging phenomenon, manifests in all aging populations. Given the rise in longevity, confronting neurodegeneration has emerged as a significant concern for healthcare frameworks, predominantly in developed nations. The rate of neuronal death progression serves as a pivotal metric for neurodegeneration. Predominant theories suggest the existence of multiple external and internal factors that can either accelerate or retard this process. Such determinants could be inherent, like an individual’s unique metabolic processes, or they might be external and associated with environmental conditions [191].

### 10.2. Potential Challenges and Areas of Unmet Need

The intricate endeavor of extrapolating specific therapeutic interventions in human-centric randomized controlled trials (RCTs) can be exemplified by neurofibromatosis type 1 (NF1). NF1 is a genetically inherited condition linked with cognitive anomalies impacting a vast majority, approximately 80%, of pediatric patients [192]. Preliminary trials have pinpointed multiple therapeutic prospects. Lovastatin rectifies synaptic functionality and ameliorates learning anomalies in Nf1+/− mice by targeting RAS activity. In contrast, compounds like methylphenidate and L-dopa bolster attention by restoring dopamine equilibrium in specific Nf1+/− variants with bi-allelic deactivation in neuroglial progenitor cells [193]. Recent reports indicate that atrophy of the basal forebrain cholinergic system (BFCS) often precedes both entorhinal cortex degeneration and memory dysfunctions in Alzheimer’s disease (AD). This revelation challenges established paradigms concerning the chronological progression of AD-associated topographical pathology [194]. Over the past decade, BACE1 has been a focal point for the development of potential AD treatments. However, formulating such compounds has proven arduous, with challenges like cellular penetration, oral absorption/metabolism, and brain accessibility. Utilizing a fragment-based chemical approach, LY2811376 ((S)-4-(2,4-difluoro-5-pyrimidin-5-yl-phenyl)-4-methyl-5,6-dihydro-4H-[1,3]thiazin-2-ylamine), the inaugural non-peptidic BACE1 inhibitor with oral bioavailability, was synthesized. This compound has demonstrated significant reductions in Aβ levels in experimental models [195].

### 10.3. Multidisciplinary Collaboration for Accelerated Biomarker Discovery

To encapsulate, a holistic methodology has been showcased, a methodology that embodies the seamless amalgamation of foundational, translational, and clinical research factions within a unified, adaptive structure centered around the NBGB. This framework consolidates clinical data, biomarkers, and post-mortem samples, coupled with comprehensive information from well-documented subjects. This integration facilitates various research teams to efficiently collate expansive and enriched data repositories, furthering investigations across multiple neurodegenerative diseases [196]. Advancing our comprehension of the intricate interplay between oxidation, antioxidants, and neurodegenerative maladies will necessitate a multidisciplinary approach [197]. Among the array of neuroimaging modalities, three techniques stand out in specialized clinical contexts due to their advanced validation stages relative to other biomarkers. These are structural MRI for atrophy detection, FDG-PET for hypometabolism assessment, and amyloid-PET for amyloid deposition quantification. In addition to these three types of MRI, neuromelanin-sensitive MRI can offer valuable information in patients with Parkinson’s disease [198,199]. The sequential application of these tools, as recommended by a consortium of multidisciplinary experts [200], draws upon their individual merits and limitations, as succinctly delineated in subsequent literature [201].

## Figures and Tables

**Figure 1 ijms-24-16119-f001:**
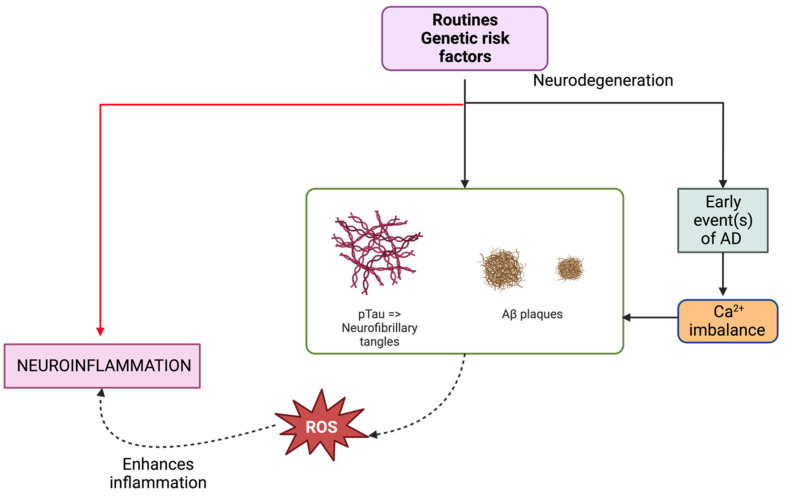
In examining the underpinnings and evolution of Alzheimer’s disease (AD), one observes substantial influence from lifestyle and risk-associated genes. A notable precursor event to AD is calcium dyshomeostasis, which plays a crucial role in the synthesis of amyloid beta (Aβ) and phosphorylated Tau (pTau). The formation of toxic oligomers from Aβ can subsequently amalgamate into amyloid plaques. Similarly, pTau oligomers can lead to the establishment of neurofibrillary tangles. Such occurrences are widely regarded as pivotal to the process of neurodegeneration. Concurrently, these and other associated phenomena generate reactive oxygen species (ROS), which not only underscore neuroinflammation, a hallmark of neurodegeneration, but also potentially exacerbate or catalyze further pathological events pertinent to AD.

**Figure 2 ijms-24-16119-f002:**
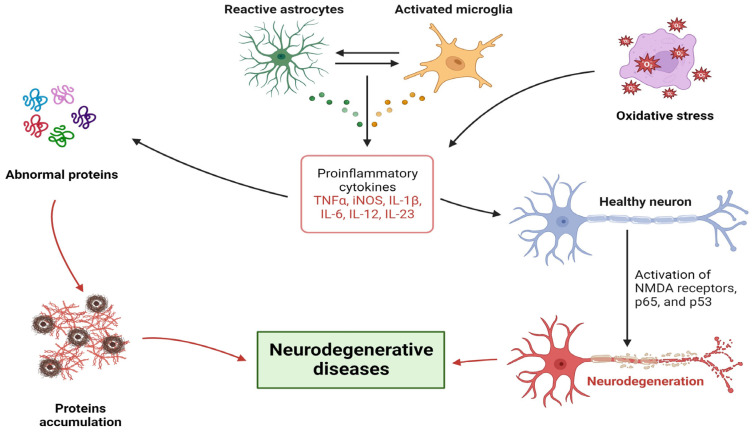
General pathophysiological mechanisms of neurodegenerative diseases.

**Table 1 ijms-24-16119-t001:** Biomarkers used for Huntington’s Disease.

References	Biomarker(s) Type/Tool	Key Findings	Implication/Significance
[38,39]	White Matter (Imaging)	Emphasis on its significance; alterations seen over a decade before anticipated disease onset.	Crucial for understanding disease progression.
[40]	Imaging Data	Data amalgamated from 1082 participants over 1–10 years from various studies.	Indicates imaging as a viable endpoint in clinical trials due to heightened sensitivity.
[41,42]	mHTT (Wet Biomarker)	Correlates with clinical scores, CSF tau, and NfL.	mHTT potentially released from compromised neurons; possible biomarker for HD.
[43]	mHTT (Wet Biomarker)	Significant in HD pathogenesis.	Crucial gauge of pharmacodynamics for huntingtin-lowering therapies.
[44]	CSF mHTT (Wet Biomarker)	Derived from striatal cells.	Suggested as PD biomarker for therapeutic engagement evaluations.
[45]	[11C]CHDI-626 (PET)	Examined for mHTT PET imaging in zQ175DN mouse model.	Proves effective for mHTT PET imaging despite rapid metabolism.
[46]	mHTT and CBVa	Early HTT-lowering treatment defers onset and decelerates progression in mHTT mouse model; CBVa alteration influenced by mHTT on neural activity.	Indicates potential therapeutic interventions and understanding neuronal dysfunction mechanisms.

**Table 2 ijms-24-16119-t002:** Biomarkers used for Multiple Sclerosis.

References	Biomarker(s)	Description/Function	Sample Origin
[47,48,49]	MRI	Used to elucidate the dimensions, quantity, chronology, and evolution of lesions in the CNS	Central Nervous System (CNS)
[50]	MIP-1a	Cytokine biomarker indicative of T cell activity	Cerebrospinal Fluid (CSF)
[50]	CXCL10	Cytokine biomarker indicative of T cell activity	Cerebrospinal Fluid (CSF)
[50]	CXCL13	Chemokine targeting B lymphocytes	Not Specified
[51,52,53,54,55]	IgG Oligoclonal Bands (OCBs)	Indicator of adaptive immunity activation in the CNS	Cerebrospinal Fluid (CSF)
[56,57,58,59]	sNfL (serum neurofilament light chain)	Potential prognostic and treatment response biomarker	Blood Serum

**Table 3 ijms-24-16119-t003:** Biomarkers used for Frontotemporal Dementia.

References	Biomarker(s)	Sample Type	Key Findings	Clinical Implications
[60,61,62,63,64]	Neurofilament light chain (NfL)	Not specified	Correlation with the pace of clinical progression. Discriminative potential between bvFTD and primary psychiatric disorders; areas under the curve: 0.84 to 0.94	Prognostic insights and discriminative utility between bvFTD and psychiatric disorders
[65,66,67]	Progranulin (GRN)	Blood, CSF	Remarkable sensitivity and specificity exceeding 95% at a threshold of 61.5 ng/mL (using AdipoGen assay). Elevated threshold proposed: 71.0 ng/mL, with a sensitivity of 98.1% and specificity of 98.5%. Levels show stability over extended periods	Potentially discriminative and diagnostic for FTLD. Stability over time suggests reliable biomarker potential

**Table 4 ijms-24-16119-t004:** Biomarkers used for Frontotemporal Dementia.

References	Biomarker(s)	Sample Type	Key Findings	Clinical Implications
[68]	Uric acid	Not specified	No significant deviations between sporadic ET, hereditary ET, and controls. A correlation between reduced uric acid levels and later age of onset in sporadic ET.	Uric acid levels may have significance as an indicator of neurodegeneration in sporadic ET patients.
[69]	Cerebellar MRI biomarkers (voxel-based morphometry, cerebellar gray and white matter volumetry, cerebellar lobular volumetry)	MRI	No cerebellar involvement identified for advanced ET across multiple MRI biomarkers and statistical models.	No significant cerebellar alterations in advanced ET.
[70]	Erythrocytic total and aggregated α-syn	Blood (erythrocyte)	Erythrocytic total and aggregated α-syn levels significantly elevated in PD and ET vs. HCs. Erythrocytic total α-syn levels higher in ET than PD. Reduced ratios of erythrocytic aggregated to total α-syn in ET vs. PD and HCs. Correlation between erythrocytic aggregated α-syn levels and disease duration in ET.	Erythrocytic α-syn concentrations might have diagnostic potential for distinguishing ET, PD, and HCs. The biomarker also correlates with ET progression.

**Table 5 ijms-24-16119-t005:** Emerging Biomarkers for Alzheimer’s Disease.

References	Biomarker(s)	Sample Type/Method	Key Findings/Notes
[71]	ADCOMS	Not specified	Primary evaluation criterion for lecanemab trial
[72]	Amyloid through PET SUVr, ADCOMS, CDR-SB, ADAS-Cog14, CSF biomarkers, Hippocampal volume (vMRI)	PET, CSF, vMRI	Secondary criteria for lecanemab trial
[74]	CTSB, BDNF, klotho	Blood (systemic)	Evaluated after a 26-week aerobic regimen
[75]	Aβ40, Aβ42/40	CSF	Dietary patterns influenced these biomarkers
[76]	GFAP	Plasma	Elevated in preclinical AD vs. healthy elderly
[84]	REST protein	CNS and in vitro	Potential novel biomarker for AD
[85]	NPTXR	Not specified	Indicative of AD progression
[86]	APOE ε4 allele	Genetic	Major determinant for AD susceptibility
[87]	NPTXR and 29 other brain-centric proteins	CSF	NPTXR levels decrease as AD progresses
[88]	Aβ42, T-tau, P-tau	CSF	Standard protocols used for assessment
[90]	YKL-40	Not specified	Marker for AD-related neuroinflammation
[95]	Desmosterol	Plasma	Reduced levels in AD patients; potential diagnostic biomarker

## Data Availability

All data are available online in libraries such as PubMed.

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
