# Peer review of "From Recognition to Remedy: The Significance of Biomarkers in Neurodegenerative Disease Pathology"

_ijms, 2023, doi:10.3390/ijms242216119_

Round 1

Reviewer 1 Report

Comments and Suggestions for Authors

This is a very interesting, detailed and important paper on biomarkers in neurodegenerative diseases.

I have some suggestion:

1. Please detail more the difference between biomarkers and surrogate markers and mention also the cases where surrogate markers have failed.

Suggest citing: Biomarkers: Further specification needed, not all biomarkers are surrogate markers. Please see and cite: Aronson JK. Biomarkers and surrogate endpoints. Br J Clin Pharmacol. 2005 May;59(5):491-4. doi: 10.1111/j.1365-2125.2005.02435.x.

Coley N, Andrieu S, Delrieu J, Voisin T, Vellas B. Biomarkers in Alzheimer's disease: not yet surrogate endpoints. Ann N Y Acad Sci. 2009 Oct;1180:119-24. doi: 10.1111/j.1749-6632.2009.04947.x

Weintraub WS, Lüscher TF, Pocock S. The perils of surrogate endpoints. Eur Heart J. 2015 Sep 1;36(33):2212-8. doi: 10.1093/eurheartj/ehv164

2. Biomarkers / Surrogate markers should be also easily performed.

Please consider the relation between inflammation, atherosclerosis and neurodegeneration and within this context the use of the surrogate marker for atherosclerosis the carotid intima media thickness cIMT. cIMT was debated for a long time as surrogate marker but recent studies suggest its validity. Please consider within this context following articles:

Zhang W, Xiao D, Mao Q, Xia H. Role of neuroinflammation in neurodegeneration development. Signal Transduct Target Ther. 2023 Jul 12;8(1):267. doi: 10.1038/s41392-023-01486-5. PMID: 37433768; PMCID: PMC10336149.

Saleh C. Carotid artery intima media thickness: a predictor of cognitive impairment? Front Biosci (Elite Ed). 2010 Jun 1;2(3):980-90. doi: 10.2741/e157.

Willeit P, Tschiderer L, Allara E, Reuber K, Seekircher L, Gao L, Liao X, Lonn E, Gerstein HC, Yusuf S, Brouwers FP, Asselbergs FW, van Gilst W, Anderssen SA, Grobbee DE, Kastelein JJP, Visseren FLJ, Ntaios G, Hatzitolios AI, Savopoulos C, Nieuwkerk PT, Stroes E, Walters M, Higgins P, Dawson J, Gresele P, Guglielmini G, Migliacci R, Ezhov M, Safarova M, Balakhonova T, Sato E, Amaha M, Nakamura T, Kapellas K, Jamieson LM, Skilton M, Blumenthal JA, Hinderliter A, Sherwood A, Smith PJ, van Agtmael MA, Reiss P, van Vonderen MGA, Kiechl S, Klingenschmid G, Sitzer M, Stehouwer CDA, Uthoff H, Zou ZY, Cunha AR, Neves MF, Witham MD, Park HW, Lee MS, Bae JH, Bernal E, Wachtell K, Kjeldsen SE, Olsen MH, Preiss D, Sattar N, Beishuizen E, Huisman MV, Espeland MA, Schmidt C, Agewall S, Ok E, AÅŸçi G, de Groot E, Grooteman MPC, Blankestijn PJ, Bots ML, Sweeting MJ, Thompson SG, Lorenz MW; PROG-IMT and the Proof-ATHERO Study Groups. Carotid Intima-Media Thickness Progression as Surrogate Marker for Cardiovascular Risk: Meta-Analysis of 119 Clinical Trials Involving 100 667 Patients. Circulation. 2020 Aug 18;142(7):621-642. doi: 10.1161/CIRCULATIONAHA.120.046361

3. You mention MRI imaging in neurodegenerative disease.

Please consider the neuromelanin sensitive MRI imaging. Please cite:

Cassidy CM, Zucca FA, Girgis RR, Baker SC, Weinstein JJ, Sharp ME, Bellei C, Valmadre A, Vanegas N, Kegeles LS, Brucato G, Kang UJ, Sulzer D, Zecca L, Abi-Dargham A, Horga G. Neuromelanin-sensitive MRI as a noninvasive proxy measure of dopamine function in the human brain. Proc Natl Acad Sci U S A. 2019 Mar 12;116(11):5108-5117. doi: 10.1073/pnas.1807983116.

4. Please consider as the text is very dense and complex:

Take-home points in bullet form

Author Response

Dear Reviewer,

Thank you for the suggestions, comments and advice!

We have followed your directions and we think the manuscript has improved after addressing accordingly to your expertise.

Our Best Regards!

Reviewer 2 Report

Comments and Suggestions for Authors

The authors summarize new progressions on biomarker identification and application in AD, PD and MSA. It is an interesting and important topic to be discussed.

Some concerns:

1, the authors should summarize biomarkers identified and its clinical implication in tables for AD, PD and MSA respectively.

2, the authors should also discuss biomarkers identified for other neurodegenerative diseases, such as HD, ALS, FTP, and ET.

Author Response

Dear Reviewer,

Thank you for the appreciation and comments.

We have followed your directions and we think the manuscript has improved after addressing accordingly to your expertise.

Our Best Regards!